# Cerebral Venous-Associated Brain Damage May Lead to Anxiety and Depression

**DOI:** 10.3390/jcm11236927

**Published:** 2022-11-24

**Authors:** Duo Lan, Siying Song, Milan Jia, Mengqi Wang, Baolian Jiao, Yunhuan Liu, Yuchuan Ding, Xunming Ji, Ran Meng

**Affiliations:** 1Department of Neurology, Xuanwu Hospital, Capital Medical University, Beijing 100053, China; 2Advanced Center of Stroke, Beijing Institute for Brain Disorders, Beijing 100053, China; 3Department of China-America Institute of Neuroscience, Xuanwu Hospital, Capital Medical University, Beijing 100053, China; 4HuaDong Hospital, Fudan University, Shanghai 200040, China; 5Department of Neurosurgery, Wayne State University School of Medicine, Detroit, MI 48201, USA

**Keywords:** cerebral venous outflow disturbance, anxiety, depression, white matter lesions, hypoperfusion

## Abstract

Background and purpose: Anxiety and depression are common in patients with Cerebral venous outflow disturbance (CVOD). Here, we aimed to explore possible mechanisms underlying this phenomenon. Methods: We enrolled patients diagnosed with imaging-confirmed CVOD, including internal jugular venous stenosis (IJVS) and cerebral venous sinus stenosis (CVSS) between 2017 and 2020. All of them had MRI/PWI scans. The Hamilton Anxiety Scale (HAMA) and 24-item Hamilton Depression Scale (HAMD) were used to evaluate the degree of anxiety and depression at the baseline and three months post-stenting. In addition, the relationships between the HAMA and HAMD scores, white matter lesions, and cerebral perfusion were analyzed using multiple logistic regressions. Results: A total of 61 CVOD patients (mean age 47.95 ± 15.26 years, 59.0% females) were enrolled in this study. Over 70% of them reported symptoms of anxiety and/or depression. Severe CVOD-related anxiety correlated with older age (*p* = 0.046) and comorbid hyperlipidemia (*p* = 0.005). Additionally, head noise, sleep disturbances, and white matter lesions (WMLs) were common risk factors for anxiety and depression (*p* < 0.05). WMLs were considered an independent risk factor for anxiety based on multiple regression analysis (*p* = 0.029). Self-contrast displayed that CVOD-related anxiety (*p* = 0.027) and depression (*p* = 0.017) scores could be corrected by stenting, as the hypoperfusion scores in the limbic lobes of patients with anxiety and depression were significantly higher than those in patients without. Conclusions: CVOD-induced hypoperfusion-mediated changes in the white matter microstructure may represent an underlying mechanism of anxiety and depression in patients with chronic CVOD.

## 1. Introduction

A high incidence of emotional disturbance, particularly anxiety and depression, has been observed in patients with Cerebral venous outflow disturbance (CVOD), including internal jugular vein stenosis (IJVS) and cerebral venous sinus stenosis (CVSS) [1]. However, the mechanisms underlying CVOD-induced anxiety and depression are still unclear. Previous studies have shown that anxiety and depression induced by chronic cerebral artery stenosis were related to the destruction of emotional processing pathways and changes in the white matter microstructure [2,3,4,5,6,7]. A previous study by our group showed that IJVS could induce cloud-like white matter lesions (WMLs) and cerebral arterial hypoperfusion [1,8]. We therefore suspected that WMLs and a hypo-perfusion might closely relate to CVOD-related anxiety and depression. Previous studies have shown that tinnitus [9,10], headaches [11,12], and sleep disturbances [13,14,15] can lead to anxiety and depression. Additionally, we found that patients with CVOD often presented with tinnitus, headaches, and sleep disturbances in addition to anxiety and depression.

More importantly, the incidence of CVOD-related anxiety and depression is supposed to get more attention in clinical settings due to the link between anxiety/depression and negative outcomes, such as suicidality [16]. As indicated in a previous study that the neurobiological factors, neuro-immunological biomarkers, brain-derived neurotrophic factors, and other neuromodulators was found in suicidal behavior, ref. [16] the CVOD-induced cerebral damage may not only cause anxiety/depression but also suicidality if the primary symptoms of anxiety/depression were not evaluated early enough.

Thus, we wanted to explore whether somatic clinical symptoms of CVOD patients were associated with anxiety and depression. Additionally, we aimed to examine, for the first time, probable mechanisms of CVOD-related anxiety and depression in patients with CVOD. We hypothesized that CVOD-related white matter lesions might be associated with a higher incidence of anxiety and depression in the population with CVOD.

## 2. Methods

### 2.1. Study Population

This single-center real-world retrospective study was conducted in Xuanwu Hospital, Capital Medical University. Patients with suspected CVOD were firstly screened in outpatient settings. They were then evaluated carefully by neuroimaging for the final confirmed diagnosis in inpatient settings. Only patients with confirmed CVOD were consecutively enrolled in this study from March 2018 to December 2020. All eligible patients signed consent forms prior to their enrolment. No healthy control group was included.

The inclusion criteria were as follows: (1) first, patients were noninvasively screened with a transcranial and extracranial echo-color Doppler (ECD) ultrasound based on the venous hemodynamic (VH) criteria. A subject was considered to be CVOD positive if more than two of five characteristics were fulfilled [17,18]. A confirmed diagnosis of CVOD was made using contrast-enhanced magnetic resonance venography (CE-MRV) of the brain and neck or digital subtraction angiography (DSA) [19,20]. IJVS and CVSS patients should not only show the local stenosis of the venous sinus or internal jugular vein but also show radiologic evidence of the presence of abnormally distorted vertebral venous plexuses and cortical venous plexus (Figure 1) as signs of decompensation. (2) Diagnoses of IJVS or/and CVSS had to be confirmed for the first time in our hospital. (3) Patients could not have any other illnesses associated with anxiety and depression, such as Parkinson’s disease, Alzheimer’s disease, post-traumatic stress disorder, etc. (4) Patients could not have a history of anxiety and depression or anti-anxiety/anti-depression medication use. (5) There were no age and gender preferences.

The exclusion criteria were as follows: (1) patients with internal carotid artery diseases including large-vessel arteriopathy, cerebral small-vessel disease, or congenital vascular malformations. (2) Patients that had incomplete clinical data. (3) Patients that did not complete the follow-up survey independently and effectively.

### 2.2. Procedures

Clinical features including participants’ age, gender, clinical symptoms/signs, presumable risk factors, and prognosis were collected. The most common symptoms of CVOD patients are sleep disturbances, headaches, tinnitus, eye discomfort, and head noises. Among the above common symptoms, the most specific symptom is head noises, which was defined as ringing, buzzing, and whooshing noises distributed throughout the brain [21,22].

All patients’ anxiety and depression symptoms were assessed using the Hamilton Anxiety Scale (HAMA) and the 24-item Hamilton Depression Scale (HAMD) at the baseline and 3-month post-treatment. The detailed HAMA and HAMD scales were presented in the previous study [23,24].

The HAMA scale was used to evaluate the severity of anxiety. Then, the subgroups were divided into three groups, including group-1 (normal, total score < 7), group-2 (mild anxiety, total score = 7–14), and group-3 (moderate/severe anxiety total score > 14). The HAMD scale was used to evaluate the severities of depression. The subgroups were divided based on the scores, including Group-i (normal, total score ≤ 8) and Group-ii (depression, total scores > 9).

We used the Patient Global Impression of Change (PGIC) score to assess the outcomes of the patients. The PGIC score is a semi-quantitated self-evaluation of the patients to the overall change in their symptoms using a 7-point scale (1 = very much improved, 2 = much improved, 3 = minimally improved, 4 = no change, 5 = minimally worse, 6 = much worse, or 7 = very much worse) [25]. According to the PGIC scores, the patients were divided into two groups: the good outcome group (PGIC scores ≤ 3) and poor outcome group (PGIC > 3).

Magnetic resonance imaging (MRI), including T1-weighted imaging, T2-weighted imaging (T2W), T2-weighted fluid-attenuated inversion recovery (T2W-FLAIR) and diffusion-weighted image (DWI), was used to identify the radiological pattern of white matter lesions (WMLs) (Figure 2) [26]. The detailed parameters of MRI were presented in the Appendix A.

The white matter lesions (WMLs) were evaluated on MRI FLAIR sequences based on Scheltans scales [26]. We previously observed that the locations of CVOD-related WMLs were mainly in the periventricular region and deep white matter, instead of in infratentorial regions and the basal ganglia, which was mainly seen in cerebral arteriostenosis-related WMLs [1,8]. Therefore, we focused on the evaluation in these two main regions of CVOD-related WMLs, with total Scheltans scores of 30 points (0 to 6 points for the periventricular region and 0 to 24 points for the deep white matter region). Then, a Scheltans score of more than 15 points was defined as the presence of CVOD-related WMLs.

Additionally, all the enrolled patients underwent contrast-enhanced magnetic resonance venography (CE-MRV) or computer tomography venography (CTV) of the brain and neck to confirm the diagnosis of IJVS or CVSS. A magnetic resonance perfusion-weighted image (MR-PWI) was used to evaluate the distribution of the hypoperfusion areas. The hypo-perfusion areas were mainly confirmed by the two MR-PWI sequences (mean transit time [MTT] and time to peak [TTP]). The MTT corresponds to the average time, in seconds, that red blood cells spend within a determinate volume of capillary circulation. The TTP is the time at which the contrast concentration reaches its maximum. Prolonged MTT and TTP were considered hypo-perfusion. A semi-quantitative analysis of cerebral perfusion was visually categorized on a scale based on chromatic aberration changes (from red to black), with a score of 0 indicating normality (black) and 5 indicating no perfusion (red) (Figure 3). The cerebral regions, including the frontal lobe, parietal lobe, occipital lobe, temporal lobe, limbic lobe, cerebellum, and basal ganglia were evaluated based on this semi-quantitative analysis.

All the aforementioned images were independently evaluated by two experienced radiologists. The radiologists were blinded to the results of the HAMA/HAMD scores and other clinical findings.

### 2.3. Statistical Analysis

The IBM SPSS Statistics version 23.0 was used to analyze all the data of this study. Data were presented as mean ± standard deviation for the continuous variables, and as counts and percentages for the categorical variables. χ^2^ tests and one-way analysis of variances were used to analyze the categorical and continuous variables, respectively. If the data did not meet the conditions of the χ^2^ test, Fisher’s exact tests were used. The Mann–Whitney U test was applied for the data that were not normally distributed. In this study, multiple logistic regressions were operated to analyze probable anxiety and depression factors and prognostic factors in the IJVS or CVSS cohorts. Additionally, a univariate logistic regression was carried out to identify the significance of each factor before multiple logistic regression analyses. We analyzed changes in the anxiety and depression scores before and after the treatment. Differences in the areas of hypoperfusion in patients with anxiety and depression compared to patients without anxiety and depression was also analyzed. A statistical significance was set at *p* < 0.05. A 95% confidence level was set with a 5% alpha error.

## 3. Results

### 3.1. Demographic Analysis

A sum of 104 patients was initially enrolled. Fifteen patients with a previous cerebral infarction, three patients with tumors, and eleven patients with anti-anxiety or anti-depression medication were excluded after the review of their medical history. Then, 14 patients failed to complete the last-time follow-up. Finally, a total of 61 eligible patients (mean age: 47.95 ± 15.39; 59.0% females) were included in the analysis, including 14 cases of isolated CVSS (23.0%), 29 cases of IJVS (47.5%), and 18 cases of CVSS combined with IJVS (29.5%). Anxiety and depression in 93.4% (57/61) of patients showed a chronic progressing process, while another 6.6% (4/61) of patients presented with a sub-acute onset. The median time of the symptoms’ onset was 5 (2–12) months. The average follow-up time after their discharge was 12.48 ± 7.77 months. The demographic features are displayed in Table 1.

Clinical symptoms and signs of the 61 enrolled patients included sleep disturbances (55.7%), headaches (55.7%), tinnitus (45.9%), eye discomfort (44.3%), head noises (44.3%), and neck discomfort (34.4%). Only 9.8% of patients experienced memory loss. According to the HAMA scores, anxiety was present in 72.1% (44/61) of patients and 75.4% (46/61) of patients had depression based on their HAMD scores. The risk factors likely related to anxiety and depression included being overweight (49.2%), having hyperlipidemia (24.6%), having a high blood pressure (HBP) (24.6%), and having suspected thyroid disorders (37.7%). Because the percentages of patients with a history of coronary artery disease (CAD) (3.3%), diabetes mellitus (DM) (3.3%), ischemic stroke (6.6%), and intracranial hemorrhage (3.3%) in this cohort were all very low, we did not evaluate them further. Additionally, 63.9% of the patients were found to have cloudy-like WMLs in their brain MR imaging. A total of 49.2% of patients had undergone an antiplatelet treatment, 50.8% of patients had undergone an anticoagulation treatment, and 16.4% of patients had been treated with endovascular stenting.

### 3.2. Risk Factors of CVOD-Associated Anxiety and Depression

Patients were divided into the following grades according to their degree of anxiety (Table 2): Grade-1, no anxiety (*n* = 17); Grade-2, borderline anxiety (*n* = 26); and Grade-3, anxiety (*n* = 18). Statistically significant between-grade differences were found in age (*p* = 0.046), sleep disturbances (*p* = 0.014), head noise (*p* = 0.046), hyperlipidemia (*p* = 0.005), cloudy-like WMLs (*p* = 0.034), and the HAMA scores (*p* = 0.001).

Specifically, Grade 3 patients were older than Grade 1–2 patients (*p* = 0.048). The incidences of sleep disturbances and hyperlipidemia were significantly higher in patients with anxiety than any of the other groups (sleep disturbance: 83.3% vs. 53.8%, *p* = 0.042; 83.3% vs. 35.3%, *p* = 0.004; hyperlipidemia: 55.6% vs. 11.5%, *p* = 0.002; 55.6% vs. 23.5%, *p* = 0.053). The percentages of head noise in both the borderline anxiety and anxiety groups were significantly higher than those in the non-anxiety group (55.6% vs. 17.6%, *p* = 0.02; 50% vs. 17.6%, *p* = 0.032).

### 3.3. CVOD-Associated Anxiety and Depression and WMLs

The most remarkable cloudy-like WMLs were mainly found in the anxiety group (*p* = 0.010). Sleep disturbances (*p* = 0.009), head noises (*p* = 0.005), and cloudy-like WMLs (*p* = 0.004) showed significant differences in the depression group. Additionally, logistic regression analysis showed a significant correlation between cloudy-like WMLs (*p* = 0.029) and anxiety. Therefore, age, sleep disturbance, head noise, hyperlipidemia, and cloudy-like WMLs may be risk factors for CVOD-related anxiety (Table 3).

The HAMD scores (*p* < 0.001) in the depression group were significantly higher than those in the non-depression group. Univariate analysis suggested that cloudy-like WMLs (*p* = 0.007), sleep disturbances (*p* = 0.013), and head noises (*p* = 0.012) could have been factors for depression. However, a further multivariate analysis showed negative results (Table 3).

### 3.4. CVOD-Associated Anxiety and Depression and Hypoperfusion

Eleven out of 61 patients (18%), including 5 cases with both borderline anxiety and depression (A–D group), and another 6 patients without anxiety and depression (Non-A–D group), underwent MR-PWI scans. Two senior radiologists who were blinded to the demographic and clinical features evaluated their MR-PWI maps. Cerebral arterial hypo-perfusions were found in all 11 cases, regardless of whether they had anxiety or depression.

Hypo-perfusions in the limbic lobe areas, including the mean transit time (MTT) (5.20 ± 0.84 vs. 3.33 ± 1.03; *p* = 0.010) and the time to peak (TTP) (5.60 ± 1.14 vs. 3.67 ± 0.52; *p* = 0.005), were more remarkable in the A–D group than in the Non-A–D group. However, the perfusion status in other areas of the brain showed no significant between-group differences (Table 4).

### 3.5. Severity of CVOD-Associated Anxiety and Depression Pre-Stenting vs. Post-Stenting

Ten out of 61 patients (mean age: 46.40 ± 15.70; 60.0% females) underwent stenting and anticoagulation combined with an antiplatelet treatment (stenting subgroup), and another 11 patients (mean age, 49.75 ± 17.50; 45.5% females), with the same demographic and clinical features, underwent only an anticoagulation combined with antiplatelet treatment (control). The amelioration of anxiety and depression, as evaluated by the score improvement from the baseline to post-treatment, was 4.36 ± 4.88 and 4.73 ± 4.56 in the stenting subgroup and 0.50 ± 1.08 and 0.70 ± 1.49 in the control group. The anxiety (*p* = 0.027) and depression (*p* = 0.017) scores improved significantly in the stenting group compared with the control group (Table 5).

## 4. Discussion

Our study inventively explored the risk factors which have contributed to CVOD-associated anxiety/depression. Moreover, the CVOD-induced hypoperfusion and CVOD-related WMLs may explain the underlying mechanism of CVOD-associated anxiety/depression.

### 4.1. Major Risk Factors Contributed to CVOD-Associated Anxiety/Depression

We found that older age, insomnia, and head noise were the top three risk factors related to CVOD-associated anxiety/depression. The relationship between aging and anxiety and depression has been shown in other studies [27,28]. However, the majority of our patients were very young, meaning aging-related anxiety and depression could be excluded. Older CVOD patients are more likely to develop anxiety/depression.

A previous study showed that sleeping disorders were significantly associated with anxiety and depression [14,29,30,31,32,33]. In our study, insomnia and late insomnia, as evaluated by the HAMA and HAMD sub-items, were also confirmed to be common symptoms in patients with CVOD, and thus may be significantly associated with anxiety and depression. Head noises, one of the most well-characterized symptoms of CVOD, were found to be associated with increased symptoms of anxiety and depression [34,35].

### 4.2. CVOD-Related WMLs

Our previous study found that high incidences of anxiety and depression and cloudy-like WMLs might be clinical and radiological features of patients with CVOD, and we also noticed that the WMLs might result from CVOD-mediated cerebral arterial microcirculation hypoperfusion [1]. This study further identified the close relationship between cloudy-like WMLs and anxiety and depression in patients with CVOD.

In the stenting subgroup, patients’ anxiety and depression symptoms showed a significant improvement after the CVOD was corrected by stenting. Additionally, the WMLs showed, accordingly, attenuations in the follow-up images (Figure 4). However, in the control subgroup, the anxiety and depression symptoms either continued or got worse. Thus, we concluded that CVOD-mediated cloudy-like WMLs might be one of the most important pathological underpinnings of anxiety and depression, even though other CVOD-related symptoms, such as headaches and head noises, might also contribute to anxiety and depression.

### 4.3. Cerebral Hypoperfusion and CVOD-Associated Anxiety and Depression

We found that cerebral hypoperfusion may be the major pathological change induced by CVOD. Further, the anxiety/depression symptoms and radiological findings of WMLs may be closely associated with cerebral hypoperfusion (Figure 5).

### 4.4. Hypoperfusion and CVOD

Previous studies about cerebral arterial diseases, such as ischemic stroke and lacunar infarctions, confirmed the relationship between cerebral hypo-perfusion, WMLs, and anxiety/depression, which involves damage to the white matter microstructure and neural networks related to anxiety and depression [36,37,38]. Patients with CVOD, either IJVS and/or CVSS, were confirmed to have the above pathological changes, all of which might explain the incidence rates of anxiety and depression (Figure 5). Our study noted CVOD-mediated cerebral arterial hypo-perfusion throughout the whole brain [39,40]. This phenomenon has also been further confirmed in this study. Hypo-perfusion was found in all 11 cases with the PWI scans, and cerebral arterial hypo-perfusions might be a risk factor for anxiety and depression. This association is likely underwritten by hemodynamic alterations and a cerebral micro-vascular structure impairment in the central and transmission pathways which are related to anxiety and depression [40].

### 4.5. Hypoperfusion and WMLs

Our study showed that the occurrence of cloudy-like WMLs was associated with the severity of anxiety and depression. In contrast to arterial-derived white matter lesions (WMLs), which present with clear boundaries and asymmetrical distributions, CVOD-induced WMLs have a bilaterally symmetric, cloudy-like appearance [1,8]. A cerebral venous obstruction may block the outflow of intracranial venous blood, and further impede the blood in capillaries reflowing into the venules, which therefore blocks the arterial blood coming into the capillaries. A long-term venous impediment from cerebral venous hypertension results in an insufficient arterial blood supply to the brain. Finally, the levels of cerebral perfusion decreased significantly. Given that CVOD affected the whole arteriovenous circulation system, CVOD-mediated hypo-perfusion was bilateral and symmetrical, rather than the focal patterns seen in arterial disease [40]. Cerebral hypoperfusion can lead to microstructural damages that are radiologically manifested as cloudy-like white matter hyperintensities. In this study, we found that stenting could improve the anxiety and depression scores. Previous studies also showed that stent therapy could improve cerebral hypoperfusion and long-term outcomes in patients with CVSS or IJVS [41,42,43]. These findings may reveal an internal link between anxiety and depression symptoms, cloudy-like WMLs, and CVOD-mediated hypo-perfusion in patients with CVOD.

### 4.6. Hypoperfusion and Damage of Neural Networks

More severe hypoperfusion can lead to a disruption of the brain’s microstructures, which in turn leads to the disruption of neural circuits, leading to anxiety and depression. A neural network impairment, such as a bidirectional projection from the orbitofrontal and medial prefrontal cortex and anterior cingulate to the amygdala and nucleus accumbens, may contribute to depression by affecting the autonomic, behavioral, and endocrine features [44,45,46,47]. In this study, the degree of limbic lobe hypoperfusion was more obvious in patients with anxiety and depression than in patients without. We observed that the areas of CVOD-induced WMLs were more widespread, [1,8] and may be more likely to involve associate conduction tracts which are known to contribute to the development of anxiety and depression.

### 4.7. Hypoperfusion and Change in Cerebral Metabolism

A positron emission tomography study on patients with generalized anxiety disorder demonstrated relative increases in the glucose metabolism in parts of the occipital, right posterior temporal lobe, inferior gyrus, cerebellum, and right frontal gyrus, and an absolute decrease in the basal ganglia [48]. Our previous study revealed that IJVS and CVSS patients showed a simultaneous hypo-perfusion and glucose metabolic status elevation [8,22,49]. Long-term hypoperfusion is implicated in the propensity of the brain tissue metabolic rates which have to increase in order to meet the basic cellular needs [50]. The abnormal metabolic pattern in patients with CVOD may correspond to the glucose metabolism patterns related to anxiety and depression, pointing to another potential pathogenic mechanism.

### 4.8. Hypoperfusion and Abnormal EEG

Previous studies showed that patients with comorbidities of anxiety and depression had abnormal EEG signals in the frontal, parietal, and insular cortices [51]. For patients with chronic cerebral ischemia (CCI), abnormal electrical activity patterns can be observed in the frontal lobe [51]. Our previous study also confirmed that CVOD could induce CCI [52]. Abnormal EEG changes caused by CCI may be associated with CVOD-associated anxiety/depression.

### 4.9. Prospects for Future Insights

Our team has several previous studies focusing on the evaluation of the difference between isolated CVSS, IJVS, and the two combined. Bai et al. assessed the clinical and imaging features of isolated CVSS, IJVS, and the combined [53]. A higher incidence of severe papilledema was more common in isolated CVSS, while IJVS patients presented with more abnormal collateral vessels. Song et al. found that different inflammatory biomarker levels presented in isolated CVSS, IJVS, and combined patients [54]. Compared with IJVS patients, CVSS patients had higher levels of the C-reactive protein. However, due to the small sample size in this study, we did not perform the subgroup analysis based on the CVOD subtypes. A further evaluation of whether there is a difference in the WML and hypoperfusion patterns between isolated CVSS, IJVS, and the combined is highly needed in a future study. Moreover, whether a longer onset process of CVOD (such as less than a 6-month onset, over a 12-month onset, etc.) may relate to more advanced symptoms/radiological changes is of our interest to further explore in studies with large sample sizes.

### 4.10. Limitations

Our study had several limitations. First, it was a retrospective single-center study with a small sample size. A further study with larger samples and more factor analyses should be carried out. Additionally, the HAMA and HAMD scores are not completely subjective assessments, and whether these assessment methods are suitable for CVOD patients with anxiety and depression is worth further discussion. A more accurate assessment system is needed in further studies.

## 5. Conclusions

CVOD-induced hypoperfusion-mediated changes in the white matter microstructure may underlie the development of anxiety and depression in patients with CVOD. CVOD patients who were eligible for intravenous stenting surgery may have a better prognosis with the relief of both somatic and affective symptoms.

## Figures and Tables

**Figure 1 jcm-11-06927-f001:**
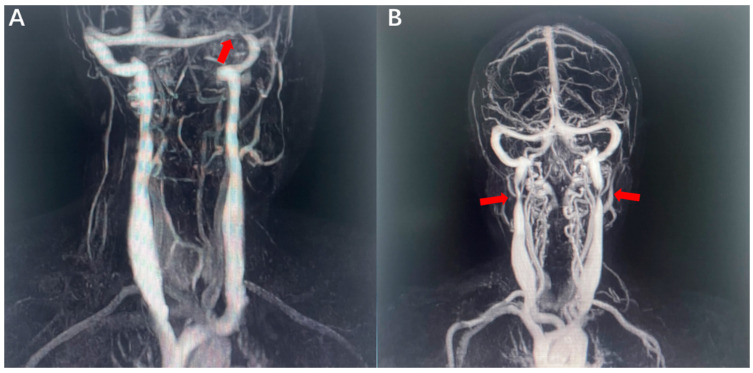
Representatives of CE-MRV findings in patients with IJVS and CVSS. (**A**) The initial CE-MRV showed severe CVSS (Red arrow), which was surrounded by abnormally distorted vertebral venous plexuses. (**B**) The initial CE-MRV showed IJVS (Red arrow) with abnormal distorted cortical and vertebral venous plexus.

**Figure 2 jcm-11-06927-f002:**
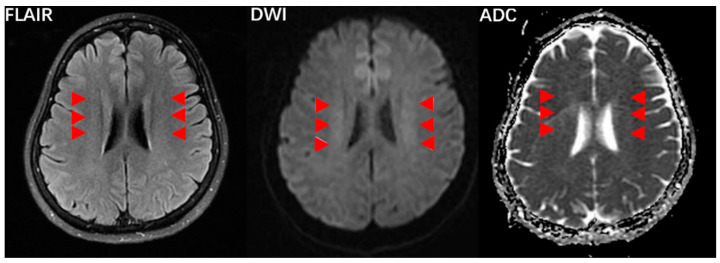
MRI findings in patients with cerebral venous outflow stenosis and IJVS conduces bilateral and symmetrical cloudy-like white matter hyperintensity (Red arrow) surrounding ventricles and centrum semiovale: T2W-FLAIR, DWI, and ADC.

**Figure 3 jcm-11-06927-f003:**
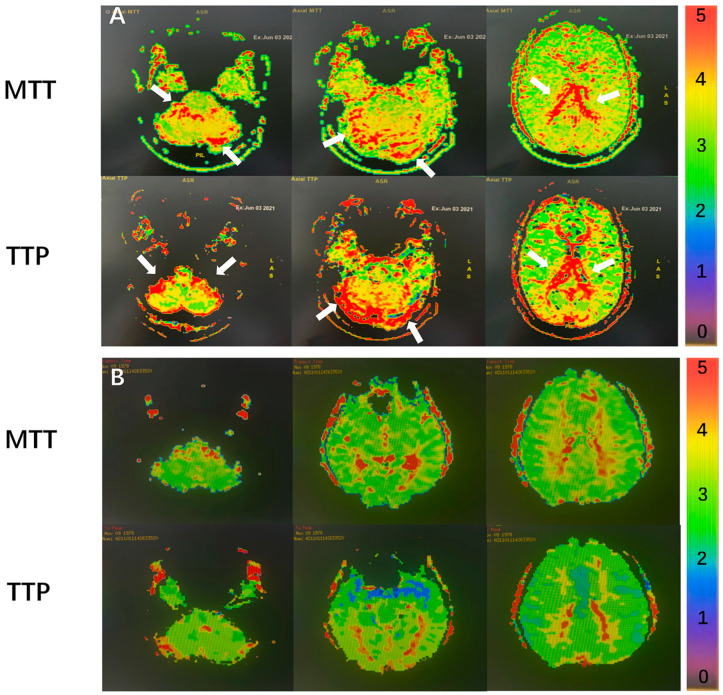
Representatives of PWI findings in patients with cerebral venous outflow stenosis and normal controls. Compared with normal controls (**B**), cerebral perfusion in patients with cerebral venous outflow stenosis (**A**) was significantly decreased. White arrows indicate the areas of hypoperfusion.

**Figure 4 jcm-11-06927-f004:**
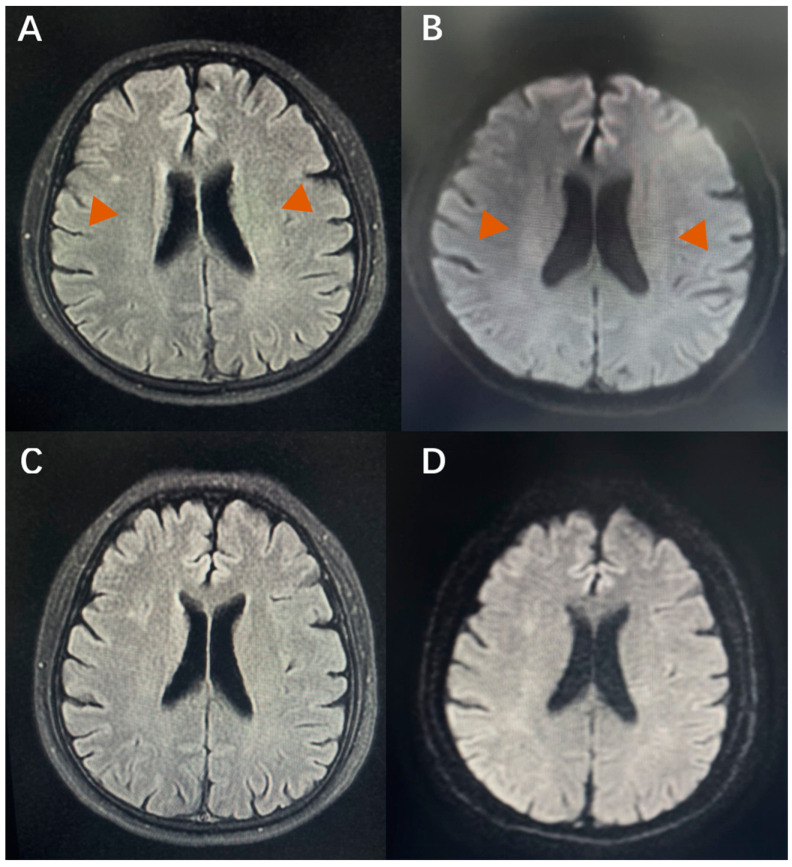
Cloudy-like WMLs (Red arrow) improved after stenting. Compared with the MR imaging before stenting (**A**,**B**), follow-up images showed attenuation in WMLs (**C**,**D**).

**Figure 5 jcm-11-06927-f005:**
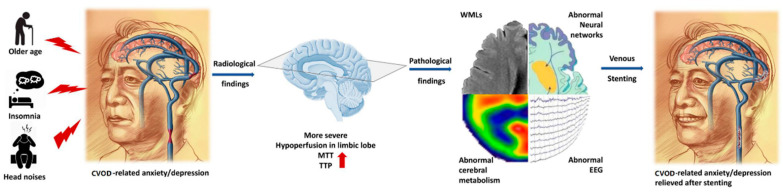
Probable mechanisms leading to anxiety and depression in CVOD patients.

**Table 1 jcm-11-06927-t001:** Demographic and basic clinical features.

Variables	All (*n* = 61)
**Personal data**	
Age, mean ± SD, years	47.95 ± 15.39
Gender (M: F)	25: 36
Course of disease	
Subacute (within 1 month)	4 (6.6%)
Chronic (more than 1 month)	57 (93.4%)
Follow-up time, months ^^^	12.48 ± 7.77
**Type of disease**	
CVSS	14 (23.0%)
IJVS	29 (47.5%)
CVSS and IJVS	18 (29.5%)
**Mist-like WMLs**	39(63.9%)
**Symptoms and signs**	
Sleep disturbances	34 (55.7%)
Eye discomfort	27 (44.3%)
Head noises	27 (44.3%)
Tinnitus	28 (45.9%)
Headache	34 (55.7%)
Neck discomfort	21 (34.4%)
Memory loss	6 (9.8%)
**Risk factors**	
Overweight (BMI > 25)	30(49.2%)
Hyperlipidemia	17 (27.9%)
HBP	15 (24.6%)
Abnormal thyroid function	23 (37.7%)
CAD	2 (3.3%)
Type 2 DM	2 (3.3%)
IS history	4 (6.6%)
ICH history	2 (3.3%)
**Treatment**	
Antiplatelet	30 (49.2%)
Anticoagulants	31 (50.8%)
Stenting	10 (16.4%)
**Anxiety ***	44(72.1%)
**Depression ^#^**	46(75.4%)
**Anxiety or depression**	46(75.4%)
**Prognosis**	
Good outcome	28 (45.9%)
Poor outcome	33 (55.1%)

^^^ Time from discharge to follow up, months. BMI = body mass index; HBP = high blood pressure; CAD = coronary artery disease; DM = diabetes mellitus; IS = ischemic stroke; ICH = intracranial hemorrhage; WMLs = white matter lesions; CVSS = cerebral venous sinus stenosis; IJVS = internal jugular vein stenosis; good outcome: PGIC ≤ 3; and poor outcome: PGIC > 3. * Including borderline anxiety and more severe anxiety. ^#^ Including borderline depression and more severe depression.

**Table 2 jcm-11-06927-t002:** Analysis of significance between CVOD-associated anxiety/depression groups.

Variables	Group I (*n* = 17)	Group II (*n* = 26)	Group III (*n* = 18)	*p* Value	Group i (*n* = 15)	Group ii (*n* = 46)	*p* Value
**Personal data**							
Age, mean ± SD	42.18 ± 17.69	47.08 ± 14.53	54.67 ± 12.18	**0.046**	42.27 ± 17.17	49.80 ± 14.49	0.100
Gender (M: F)	9: 8	10: 16	6: 12	0.470	8: 7	17: 29	0.263
Course of disease				NA			1.000
Subacute	3 (17.6%)	1(3.8%)	0 (0%)		1 (6.7%)	3 (6.5%)	
Chronic	14 (82.4%)	25 (96.2%)	18 (100%)		14 (93.3%)	43 (93.5%)	
Follow-up time ^	13.76 ± 8.00	10.81 ± 7.13	13.67 ± 8.60	0.364	13.33 ± 7.84	12.20 ± 7.90	0.629
**Type of disease**				0.201			0.391
CVSS	5 (29.4%)	8 (30.8%)	1 (5.6%)		5 (33.3%)	9 (19.6%)	
IJVS	6 (35.3%)	11 (42.3%)	12 (66.7%)		5 (33.3%)	24 (52.2%)	
CVSS and IJVS	6 (35.3%)	7 (26.9%)	5 (27.8%)		5 (33.3%)	13 (28.3%)	
**Cloudy-like WMLs**	7 (41.2%)	17 (65.4%)	15 (83.3%)	**0.034**	5 (33.3%)	34 (73.9%)	**0.004**
**Clinical symptoms**							
Sleep disturbance	6 (35.3%)	14 (53.8%)	15 (83.3%)	**0.014**	4 (26.7%)	30 (65.2%)	**0.009**
Eye discomfort	8 (47.1%)	9 (34.6%)	10 (55.6%)	0.394	6 (40.0%)	21 (45.7%)	0.702
Head noise	3 (17.6%)	13 (50%)	10 (55.6%)	**0.046**	2 (13.3%)	25 (54.3%)	**0.005**
Tinnitus	6 (35.3%)	14 (53.8%)	8 (44.4%)	0.485	7 (46.7%)	21 (45.7%)	0.945
Headache	8 (47.1%)	16 (61.5%)	10 (55.6%)	0.647	6 (40.0%)	29 (63.0%)	0.117
Neck discomfort	7 (41.2%)	8 (30.8%)	6 (33.3%)	0.776	5 (33.3%)	16 (34.8%)	0.918
Memory loss	1 (5.9%)	3 (11.5%)	2 (11.1%)	1.000	1 (6.7%)	5 (10.9%)	1.000
**Risk factors**							
Overweight	8 (47.1%)	12 (46.2%)	9 (50%)	1.000	7 (46.7%)	21 (45.7%)	0.945
Hyperlipidemia	4 (23.5%)	3 (11.5%)	10 (55.6%)	**0.005**	2 (13.3%)	15 (32.6%)	0.196
HBP	3 (17.6%)	6 (23.1%)	6 (33.3%)	0.564	3 (20.0%)	12 (26.1%)	0.634
Abnormal thyroid	6 (35.3%)	10 (38.5%)	7 (38.9%)	0.971	5 (33.3%)	18 (39.1%)	0.687
**Treatment**							
Antiplatelet	6 (35.3%)	14 (53.8%)	10 (55.6%)	0.400	7 (46.7%)	23 (50.0%)	0.823
Anticoagulant	11 (64.7%)	14 (53.8%)	6 (33.3%)	0.173	11 (73.3%)	20 (43.5%)	0.073
Stenting	4 (23.5%)	4 (15.4%)	2 (11.1%)	0.682	4 (26.7%)	6 (13.0%)	0.216
**Score * (mean ± SD)**	3.18 ± 1.91	10.77 ± 1.95	20.61 ± 6.75	**<0.001**	3.47 ± 2.23	16.11 ± 7.227	**<0.001**

Group I = without anxiety; Group II = borderline anxiety; Group III = certain anxiety; Group i = without depression; Group ii = depression; CVSS = cerebral venous sinus stenosis; IJVS = internal jugular vein stenosis; HBP = high blood pressure; and WMLs = white matter lesions. ^ Time from discharge to follow up, months. * The score of the Hamilton Anxiety Scale (HAMA) or Hamilton Depression Scale (HAMD).

**Table 3 jcm-11-06927-t003:** Univariate and Multivariate logistic regression analysis between depression and risk factors.

Variable	Univariate ^#^	*p* Value *	Multivariate ^#^	*p* Value *
**Depression**				
Age	1.03(0.99, 1.07)	0.104	-	-
Sleep disturbances	0.19(0.05, 0.71)	**0.013**	0.30(0.07, 1.23)	0.095
Head noises	0.13(0.03, 0.64)	**0.012**	0.24(0.04, 1.29)	0.096
White matter lesion	0.18(0.05, 0.62)	**0.007**	0.30(0.08,1.20)	0.089
**Anxiety**				
Age	0.02(0.01, 0.07)	**0.015**	1.02(0.98, 1.07)	0.248
Sleep disturbances	0.31(0.10, 1.00)	0.051	0.41(0.12, 1.43)	0.162
Head noises	1.83(0.58, 5.83)	0.305	-	-
Hyperlipidemia	0.73(0.20, 2.68)	0.054	-	-
White matter lesion	0.26(0.08, 0.85)	**0.025**	3.99(1.15, 13.8)	**0.029**

^#^ Results were presented as odds ratio (95% CI). * Statistically significant at *p* < 0.05.

**Table 4 jcm-11-06927-t004:** The difference in hypoperfusion scores in different lobes of the brain between the anxiety and depression group and the non-anxiety and depression group.

Scores (Mean ± SD)	MTT		*p* Value	TTP		*p* Value
	A–D Group	Non-A–D Group		A–D Group	Non-A–D Group	
**Frontal lobe**	4.60 ± 1.52	5.50 ± 2.74	0.530	5.40 ± 1.67	4.00 ± 1.67	0.200
**Parietal lobe**	4.40 ± 0.89	4.00 ± 1.67	0.644	4.60 ± 1.67	4.00 ± 1.27	0.515
**Occipital lobe**	6.00 ± 2.00	5.50 ± 1.38	0.635	5.60 ± 1.82	5.33 ± 1.97	0.822
**Temporal lobe**	4.60 ± 1.52	3.83 ± 1.72	0.458	4.00 ± 0.71	3.83 ± 1.83	0.853
**Limbic lobe**	5.20 ± 0.84	3.33 ± 1.03	**0.010**	5.60 ± 1.14	3.67 ± 0.52	**0.005**
**Cerebellum**	4.00 ± 1.73	5.00 ± 1.67	0.357	3.60 ± 1.14	4.67 ± 2.34	0.378
**Basal ganglia**	3.40 ± 0.89	3.00 ± 1.27	0.568	3.80 ± 0.84	3.83 ± 0.75	0.946

MTT: mean transit time; TTP: time to peak; A–D group: patients with anxiety and depression; and non-A–D group: patients without anxiety and depression.

**Table 5 jcm-11-06927-t005:** Comparison of anxiety and depression scores between the stent group and the control group at 3 months after treatment.

Scores (Mean ± SD)	Anxiety			Depression		
	Group S	Group C	*p*	Group S	Group C	*p*
At admission	10.73 ± 10.15	9.50 ± 7.37	0.757	12.45 ± 10.27	8.40 ± 6.54	0.300
Three months after treatment	6.45 ± 5.65	9.00 ± 6.78	0.360	7.73 ± 6.47	7.70 ± 6.06	0.992
Difference	4.36 ± 4.88	0.50 ± 1.08	**0.027**	4.73 ± 4.56	0.70 ± 1.49	**0.017**

Group S: the patients underwent stent implantation on the basis of anticoagulation combined with antiplatelet therapy. Group C: this group of patients received antiplatelet combined with anticoagulation therapy.

## Data Availability

The data and materials are available on request to the corresponding author.

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
