# Peer review of "Cerebral Venous-Associated Brain Damage May Lead to Anxiety and Depression"

_jcm, 2022, doi:10.3390/jcm11236927_

Round 1

Reviewer 1 Report

Congratulations on this well-rounded, inventive and nicely executed study. Very interesting results showing the association of CVOD, white matter lesions and anxiety and depression. Good combination of clinical variables with imaging and psychiatric assessments, nicely drawn links, well-explained reasoning and arguments, thorough discussion. It is indeed a pity that this study is relatively small, retrospective and single-center. Results from a larger and longer study would certainly be of high interest. I have a few minor remarks and questions.  

In the sentence: “Previous studies have shown that anxiety and depression induced by chronic cerebral artery stenosis were related to the destruction of emotional processing pathways and changes in white matter microstructure” you do not provide any references – is this on purpose or are they missing?

Could you please provide the numbers for selection of patients – i.e. how many patients were excluded for which reasons? I would be interested to know for instance how many patients were excluded due to prior history of depression and anxiety and how many are lost in follow-up.

Could you also please clarify what is meant by “illnesses associated with anxiety and depression” – I assume you also excluded patients with a history of depression and anxiety?

Thank you for a very clear figure 1. Could you please specify how the diagnosis of CVOD was confirmed using MRV or DSA – which criteria/thresholds were used? As far as I can see the reference you use is for ultrasound diagnosis?

Could you please explain what specifically is meant by prolonged MTT and TTP? How is prolonged defined?

You indicate that two senior radiologists assessed all images – does it mean that all images were assessed twice – independently and blinded by two radiologists – or were the images divided and some were assessed by one and some by the other radiologist? Could you please indicate whether the radiologists assessing the hypoperfusion/WMLs were blinded to the results of HAMA/HAMD scores and other clinical findings?

Figure 3 indicates PWI findings in CVOD patients and in controls. Clear figure – but just to clarify – in your study you in fact did not have an imaging control group to compare with, correct? The methods section do not mention any control groups.

Were there differences between patients with isolated CVSS, IJVS and combined? Could you please hypothesize whether studying these groups separately, if the numbers allow – would be of interest? Could they show different WNL and hypoperfusion patterns and lead to different outcomes?

More than 93% of patients showed a chronic onset (>1 month), if possible, could you indicate a median (IQR) of how long it was in the included patients? Could it be different for patients who have different onset times – e.g. patients who have 2 month onset would be different to those who have 12 month onset process as the latter could suffer from more advanced symptoms/radiological changes?

I would not express follow-up months as 12.48 +/- 7.77 – it is unclear to me how much is 0.77 of a month… Similar with age of the included patients – why did you decide to express age until 2 decimal places?

In Table 1 you indicate poor/good prognosis – could you please clarify what is meant by this?

In Table 3 you present negative odds ratios. Although I am not a statistician I am curious how you arrived at those.

Could you please refer more thoroughly to possible confounders in your study? Could it be that other symptoms (such as head noise or sleep disturbances or comorbidities) symptoms are confounders associated with both WNLs and psychiatric symptoms? What was the influence of different treatments?

Could you please hypothesize if your findings have application in cerebral venous thrombosis patients?

Reviewer 2 Report

In general, the article seems to be well documented, with interesting conclusions. I would suggest to insert the following bibliographic titles related to white matter changes in depression and the potential effect of anticoagulant and antidepressant treatment on the evolution of cerebral venous thrombosis and the influence on the white matter microstructure. 

1. He E, Liu M, Gong S, Fu X, Han Y, Deng F. White Matter Alterations in Depressive Disorder. Front Immunol. 2022 May 12;13:826812. doi: 10.3389/fimmu.2022.826812. PMID: 35634314; PMCID: PMC9133348.

2. Seiger R, Gryglewski G, Klöbl M, Kautzky A, Godbersen GM, Rischka L, Vanicek T, Hienert M, Unterholzner J, Silberbauer LR, Michenthaler P, Handschuh P, Hahn A, Kasper S, Lanzenberger R. The Influence of Acute SSRI Administration on White Matter Microstructure in Patients Suffering From Major Depressive Disorder and Healthy Controls. Int J Neuropsychopharmacol. 2021 Jul 23;24(7):542-550. doi: 10.1093/ijnp/pyab008. PMID: 33667309; PMCID: PMC8299824.

3. Bajko Z, Maier S, Motataianu A, Filep RC, Stoian A, Andone S, Balasa R. Rivaroxaban for the treatment of cerebral venous thrombosis: a single-center experience. Acta Neurol Belg. 2022 Feb;122(1):105-111. doi: 10.1007/s13760-021-01651-z. Epub 2021 Mar 18. PMID: 33733345.

Author Response

Thanks so much for your valuable comments. We have carefully revised our manuscript based on your kind suggestion. All your suggested references were added in the related parts.